# The Glyoxalase System—New Insights into an Ancient Metabolism

**DOI:** 10.3390/antiox9100939

**Published:** 2020-10-01

**Authors:** Jakob Morgenstern, Marta Campos Campos, Peter Nawroth, Thomas Fleming

**Affiliations:** 1Department of Internal Medicine I and Clinical Chemistry, University Hospital Heidelberg, 69120 Heidelberg, Germany; Marta.Campos@med.uni-heidelberg.de (M.C.C.); Peter.nawroth@med.uni-heidelberg.de (P.N.); thomas.fleming@med.uni-heidelberg.de (T.F.); 2German Center for Diabetes Research (DZD), 85764 Neuherberg, Germany; 3Institute for Diabetes and Cancer at Helmholtz Zentrum Munich, 85764 Neuherberg, Germany

**Keywords:** Glyoxalase 1, methylglyoxal, glycation

## Abstract

The glyoxalase system was discovered over a hundred years ago and since then it has been claimed to provide the role of an indispensable enzyme system in order to protect cells from a toxic byproduct of glycolysis. This review gives a broad overview of what has been postulated in the last 30 years of glyoxalase research, but within this context it also challenges the concept that the glyoxalase system is an exclusive tool of detoxification and that its substrate, methylglyoxal, is solely a detrimental burden for every living cell due to its toxicity. An overview of consequences of a complete loss of the glyoxalase system in various model organisms is presented with an emphasis on the role of alternative detoxification pathways of methylglyoxal. Furthermore, this review focuses on the overlooked posttranslational modification of Glyoxalase 1 and its possible implications for cellular maintenance under various (patho-)physiological conditions. As a final note, an intriguing point of view for the substrate methylglyoxal is offered, the concept of methylglyoxal (MG)-mediated hormesis.

## 1. Introduction

The glyoxalase system, which was first described over a hundred years ago, consists of two cooperating enzymes named Glyoxalase 1 (Glo1) and Glyoxalase 2 (Glo2) [1,2]. It is a highly conserved enzyme system, which exists in every living cell from crude single prokaryotes up to complex mammalian organisms [3,4]. The initial enzyme of the glyoxalase system is glyoxalase 1 and its substrate, hemithioacetal, is formed by a spontaneous reaction of methylglyoxal (MG) and glutathione (GSH) (Figure 1). MG is highly reactive and therefore a toxic compound, but also an inevitable by-product of glycolysis and gluconeogenesis during the conversion of triose phosphate isomers (dihydroxyacetone phosphate and glyceraldehyde 3-phosphate) [5]. Interestingly, the glyoxalase system has an extremely narrow substrate specificity among dicarbonyl metabolites. In fact, it detoxifies mostly MG via the hemithioacetal, which is converted to S-D-lactoylglutathione and then hydrolyzed by Glo2 to D-lactate (Figure 2). Within this context, Glo1 represents the rate-limiting step for the detoxification of MG and is therefore of highest interest because it is believed to maintain the intracellular concentration of harmful MG in a low range. As the prevention of increased intracellular MG concentration is mandatory for the viability of a cell, in order to maintain physiological functions, it is no surprise that in eukaryotic organisms, the abundancy of Glo1 is in the top 10% of cytosolic proteins [6,7].

Glo2 is encoded by the hydroxyacylglutathione hydrolase gene and, like Glo1, it is expressed in nearly all living cells in order to hydrolyze S-D-Lactoylglutathione, the intermediate product of the glyoxalase cycle, into D-Lactate and (recycled) GSH [8,9]. Interestingly, S-D-Lactoylglutathione has been reported to be potentially cytotoxic, inducing growth arrest, and decreasing cellular viability, but further studies are needed to confirm this, because of the non-physiological doses used in this study [10]. In comparison to Glo1, Glo2 has not been located solely in the cytosol, but also in the mitochondria [8]. The amount of research conducted and therefore, available literature regarding Glo2 is far less compared to its counterpart Glo1. This may be the result of the fact that Glo1 is the rate-limiting enzyme of the glyoxalase system and seems to be more important [6]. Nevertheless, it is surprising that Glo2 is regulated by members of the tumor suppressor gene p53. In fact, Glo2 is up-regulated by p63 and p73, which are members of the p53 family, and therefore Glo2 is believed to have a role as a pro-survival factor [11]. Fortunately, recent investigations about the role of mitochondrial Glo2 have revealed new and very interesting content. Antognelli et al. suggested a pro-apoptotic role of Glo2 in non-small-cell lung cancers, and this effect is mediated by a bioactive plant compound called Oleuropein. Another study from the same group revealed that Glo2 can be viewed as a driving force of prostate tumorigenesis and therefore, may represent a novel marker of progression in prostate cancer diagnosis [12,13]. It is a nice illustration, showing that Glo2 may have been overlooked in the context of malignancies and that there is still a lot of space for further studies.

Increased levels of MG can lead to an intracellular accumulation of advanced glycation end products (AGEs) and to increased amounts of glycated DNA adducts. Both events are associated with many pathological events in humans, such as obesity and diabetes, atherosclerosis, various cancer types and neurodegenerative diseases [14,15,16,17,18,19,20]. Regarding protein glycation, the preferred targets of MG are the amino acids arginine and, to a much lesser extent, lysine. This mainly results in MG-derived hydroimidazolones called MG-H1, MG-H2 and MG-H3 (structure displayed in Figure 1). Based upon quantitative measurements, MG-H1 is the dominant type of all AGEs found within a physiological context [21,22,23]. Two other modifications should be mentioned herein, the arginine modification argpyrimidine and the most abundant lysine modification *N*-(1-carboxyethyl)lysine (CEL) [24,25]. Regarding the modifications of nucleic acids the most reactive nucleotide is deoxyguanosine, resulting in N^2^-carboxyethyl-2′-deoxyguanosine (CedG) and 3-(2′-deoxyribosyl)-6,7-dihydro-6,7-dihydroxy-6/7-methylimidazo-[2,3-b]purin-9(8)one (MG-dG) [26]. CedG has been found to be elevated in animal models and in human tissue samples, especially within the context of diabetes, but further studies are still needed to confirm its relevance (Figure 1) [27].

The regulation of Glo1 activity and expression is complex and still not well understood. However, what we know is that Glo1 has a metal responsive element, an insulin-responsive element, an antioxidant responsive element, and it is a hotspot of copy number variation [28,29,30]. Downregulation of Glo1 via hyperglycemic or hypoxic conditions induces MG stress, whereas an upregulation via Nrf2 increases Glo1 activity and therefore can alleviate intracellular MG stress [29,31]. Many experimental data suggest that the modulation of Glo1 has a high impact towards the phenotype of healthy cells exposed to cellular stress, and also of malignant cells with an increased energy demand [7,15,17,32,33,34,35]. In cruder model organisms for instance, such as *Caenorhabditis elegans* (C. elegans), an overexpression of Glo1 results in an increase in mean and maximum lifespan by ~40% [36]. In bovine endothelial cells, the overexpression of Glo1 reduces the intracellular accumulation of AGEs under hyperglycemic conditions [37]. Consistently, overexpression in rat models of diabetes was reported to be protective regarding ateriogenesis and renal impairment driven by microvascular alterations [35,38]. On the other hand, the partial loss of Glo1 is associated with detrimental kidney damage in a murine Glo1 knock-down model [39]. Hyperglycemic episodes and therefore diabetes related late complications, such as nephro-, retino- and neuropathy, have been frequently linked to increased MG levels due to a lower Glo1 expression in humans [14,17,18,40]. Furthermore, in a recent clinical trial, a Glo1 inducer (resveratrol-hesperetin) improved metabolic and vascular health in a small cohort of overweight and obese humans [41]. Last but not least, the naturally occurring decrease in Glo1 during aging seems to be associated with an increased risk of age-related cardiovascular diseases [42].

As already mentioned, in many types of cancer the situation is quite different. Due to the vast amounts of glucose consumption resulting from the increased proliferation rates of most tumors, Glo1 is necessary to protect the cell from harmful MG. In malignant cells with a high proliferation rate, it is undoubtedly proven that Glo1 protects those cells from the increased formation of MG due to a high glycolytic flux within the context of the Warburg effect [43,44,45]. Therefore, it is no surprise that Glo1 is generally overexpressed in numerous cancers potentially as a part of a survival strategy [15]. This is the case in urological malignancies, different breast cancer types, gastric cancer cells, bladder, colon and hepatocellular carcinomas as well as leukemia [46,47,48,49,50,51,52,53]

In prostate cancer, Glo1 is also linked to the maintenance of the metastatic phenotype by controlling the epithelial to mesenchymal transition [54]. The association of Glo1 with tumor growth could suggest its role as an oncogene, but this seems to be more an adaption to protect the tumor proteome against an increased intracellular flux of MG formation [55]. The inhibition of Glo1, therefore, may represent a potential target for anticancer therapeutics, although compounds such as bromobenzylglutathione cyclopentyl ester, which has been used in vitro, have to prove their therapeutic capacity in humans also [56].

This review will not focus on the past, where most of the studies have shown that the high global abundance of Glo1 is a result of a necessity to detoxify MG and prevent accumulation of AGEs and DNA modifications. This review will summarize the importance of alternative detoxification mechanisms for MG, spotlight the overlooked post-translational modifications (PTMs) of Glo1, and emphasize antioxidant aspects of MG-derived AGEs (e.g., MG-H3) and the hormetic potential of MG itself. Herein, we put very recent findings into new contexts of MG-derived damage under (patho-) physiological conditions and this will hopefully leave the reader with a challenging and eye-opening new perspective of the glyoxalase system.

## 2. Glo1 Is Not Indispensable for Crude and Complex Organisms

Due to its highly conserved role, the glyoxalase system has been viewed as a crucial enzyme system to maintain cellular viability [6,8,14]. This is a consequence of the necessity to detoxify the highly reactive dicarbonyl MG as fast as possible, in order to prevent glycation of proteins and DNA [7]. This has been proven in plants, crude prokaryotic microorganisms and many animals such as nematodes, arthropods, as well as chordates [4,5,36,57,58,59]. Interestingly, in *Escherichia coli* it has been shown that there is also a glyoxalase-like enzyme (glyoxalase 3) that is independent of any cofactor, converting MG into D-Lactate [60]. When it comes to mammalian organisms, the detoxification of MG is mainly performed by the glyoxalase system, but also other pathways of MG detoxification have been described in vitro. This includes enzymes of the aldo–keto reductase (AKR) and aldehyde dehydrogenase (ALDH) family [61,62]. Another enzyme, Deglycase DJ-1, also known as Parkinson disease protein 7, has been claimed to convert MG into lactate without the need of GSH, but the results of this recently found the enzyme point more towards a tool which can repair MG-modified nucleotides and proteins [63,64] The significance of AKR-dependent detoxification of MG has previously been shown for the first time in an in vivo mouse study. The loss of AKR1b3 in apoE knock-out (KO) mice increased the accumulation of AGEs and also the formation of atherosclerotic lesions [65]. However, when compared with a Glo1 knock-down mouse on the same apoE background, it did not show any differences in either MG-derived AGEs, such as MG-H1, or enhanced formation of atherosclerosis [66].

Unfortunately, the first attempts to abolish Glo1 activity failed, due to a high amount of copy number variants using the approach of gene trapping mutations [67]. Nevertheless, using the same approach, the first global Glo1 KO in mice was established in 2017 [68]. Recent advancements regarding genetic engineering of model organisms and isolated cells via CRISPR Cas9 technique unlocked the possibilities to generate gene KOs in a fast, convenient, and efficient way [69]. Using this CRISPR Cas9 technique, independent work groups were able to establish Glo1 KOs in zebrafish, drosophila fly, and mammalian in vitro and in vivo models [70,71,72]. Surprisingly, the effect of the complete global Glo1 KO was rather mild, at least in fish and mammals. The study in *Drosophila* revealed that Glo1 KO flies first become hyperinsulinemic in a young state, develop elevated total-body triglyceride with normal circulating glucose levels once they are matured, and then acquire hyperglycemia when they are in a mature state. The flies showed early on mildly increased MG and MG-H1 levels. This led to the authors’ hypothesis, that MG is not the cause of late diabetic complications due to increased formation of AGEs and DNA modifications, but rather more the reason for insulin resistance due to hyperglycemia similar to what is observed in type 2 diabetic patients [71]. In the model organism zebrafish, a partial knock-down of Glo1 resulted in the elevation of MG levels and in alterations of the intersomitic blood vessels, but the complete loss of Glo1 in zebrafish showed a completely different picture [57,70]. The Glo1 KO fish survived until adulthood without any obvious growth deficits and showed only mildly increased MG levels in the tissue. Once the Glo1 KO zebrafish were challenged by high nutrient intake, fasting blood glucose levels were increased and retinal blood vessel alterations occurred. In this study, it has been revealed that the relatively mild increase in MG levels may be explained by the increased expression and activity of several subtypes of the ALDH family (3a1, 3a2a, 9a1a2 and 9a1b), potentially responsible for the alternative detoxification of MG [70]. Looking into the mammalian model systems, a Glo1 KO in HEK293 cells showed elevated MG-H1 following exogenous MG stimulation as compared to wild-type cells [64]. However, under basal conditions no elevations of MG-H1 was found, reflecting a hypersensitive model for exogenously added MG, which has also been confirmed in neuronal Schwann cells with a complete loss of Glo1 [72]. In this neuronal model system, the complete loss of Glo1 resulted in a highly efficient compensation driven by AKR1b3. It is interesting to point out that it is not MG itself that is converted into a nontoxic compound, but rather the hemithioacetal, the spontaneous product of MG and GSH, that is detoxified mainly to lactaldehyde. This makes sense given the fast and effective reaction of MG with GSH in order to neutralize the highly reactive capacity of MG [72]. Previously, the enzyme AKR1b3 (in humans the isotype is called AKR1b1) was viewed only as the driving enzyme of the sorbitol pathway, which has a negative impact on the outcome of diabetic neuropathy [14]. Consequently, and based upon the in vitro findings, the inhibition of AKR1b3, which has a supporting role in the detoxification of MG, could explain why the clinical outcome for treating diabetic neuropathy with AKR inhibitors, such as epalrestat, resulted in disappointing outcomes [73]. Obviously, compensatory pathways for the glyoxalase system were underestimated in earlier studies. This is supported by the findings that in mammalian models a complete loss of Glo1 does not result in basal MG and MG-H1 elevations, even under high glucose conditions [72,74]. It has to be pointed out that most of the in vitro studies about MG-derived AGEs and DNA adducts had to use exogenously added MG in a supraphysiological range in order to achieve obvious molecular effects [64,72,75,76,77]. Researchers in the field studying MG-derived AGEs and DNA adducts may misjudge the potential of artefacts in cells, cultured under isolated conditions, and that those results cannot be linked directly to the complex physiology of an organism (see below). Certainly, such Glo1 KO studies can be contradictory and provoking to the scientific dogma that silencing or inhibiting Glo1 has detrimental effects. Previously, this merged into the assumption of a cause-and-effect model between Glo1 activity and the levels of MG. If this is true, diseases associated with a complete inherited loss of Glo1 and therefore increased MG levels would have been described. As of yet, no known diseases have been attributed to Glo1 mutations in humans.

In a mouse models with a global Glo1 KO, others were able to show that the expected damaging effects of Glo1 loss are rather mild or non-existent [68,74]. One group postulated increased levels of MG-H1 due to Glo1 loss, using an obsolete immunoblot technique, but only found this increase in the liver and not in the brain. When the loss of Glo1 in a mouse model was combined with a streptozotocin treatment in order to enhance the glycemic burden, it was revealed that organ specific compensatory pathways could substitute the role of Glo1 and detoxify vast amounts of MG into hydroxyacetone and lactaldehyde (through AKR) in the liver and kidney, and into pyruvate (through ALDH) in the liver (Figure 1) [74]. One reason for the development of late diabetic complications is the increase in MG-derived AGEs and DNA modifications, which is mainly due to the fact that Glo1 is downregulated in diabetes. Consequently, MG cannot be detoxified efficiently [14,20,21,34,77]. If this were true, then a diabetic Glo1 KO model would reflect a heavy damage of tissues which are prone to hyperglycemic conditions, such as kidneys, nerves and the eyes. However, the recent data described above, support the idea that the development of late diabetic complications due to increased AGEs and MG-derived DNA damage cannot be a result of one single downregulated enzyme. The data suggest that it is a multifactorial enzymatic system of high complexity, which is responsible for the detoxification of increased amounts of MG. Interestingly, this idea is supported by data from a small human cohort. In patients diagnosed with type 2 diabetes, but without major complications, the alternative MG detoxification product of AKR, hydroxyacetone, was elevated. In plasma of diabetic patients with complications, such as nephropathy, hydroxyacetone was not increased [74]. However, this association study needs to be proven in a larger, well-controlled study. Furthermore, due to the fact that products of MG detoxification are converted rapidly to advanced metabolites, more studies should try to approach relative flux measurements with labeled MG and the reliable determination of possible products of MG detoxification, such as D-lactate, lactaldehyde, hydroxyacetone and pyruvate in a more profound manner. Therefore, the combination of metabolic, enzymatic and genetic factors may provide a better approach to identify those patients which are at risk for the development of late diabetic complications [74].

In summary, and to put the new findings into context, various in vivo studies of Glo1 KO models show that Glo1 is not an indispensable enzyme and thus, the glyoxalase system is neither. It is also a fascinating example that with increasing complexity and advanced evolution, nature may establish a backup system, which can compensate and substitute genetic mutations of highly conserved enzymes. This may not be perfectly the case in crude organisms, such as yeast and worm, but to some degree in flies and fish, and finally with perfection in a mammalian organism (Figure 2). This leads consequently to the question of why the glyoxalase system is not mandatory for a complex organism, despite it being a highly conserved and abundant system. The next part of this review tries to find answers to that essential question.

## 3. Modifications of Glo1 and Other Possible Intracellular Implications

Despite the extensive research on the glyoxalase system, and in particular Glo1, investigating PTMs of Glo1 in mammals have not been a focus in the field. Studies regarding the PTMs of Glo1 are rare and the consequences of those are unfortunately not well understood [77]. However, PTMs of Glo1 have been described in yeast and plants, which suggest that such regulatory mechanisms are as highly conserved as the glyoxalase system itself. In *Arabidopsis thaliana*, for instance, a phosphorylation of Glo1 was observed within the context of enhanced stress resistance [78]. Furthermore, in rice leaf sheath, gibberellin, a regulator of growth, also induces phosphorylation of Glo1 in addtion to malate dehydrogenase and glyceraldehydes-3-phosphate dehydrogenase [79]. Plants in general are well characterized regarding the regulation of Glo1, where glyoxalases exist as a multigene family, and thereby reveal interesting roles beyond the view of simple MG detoxification. Some are the associations of Glo1 and cell division or that its enzyme activity is essential for compatible pollination in plants [3,80]. Furthermore, it has been revealed that MG at low levels plays a role as a signal molecule in cross-communication with jasmonic acid in *Arabidopsis thaliana* [81].

Yeast, which was exposed to the mating factor, also induced the phosphorylation of Glo1, whereas S-nitrosoglutathione inactivated Glo1. Further studies identified a nitric oxide responsive element at a cysteine residue (Cys^139^) even within the human Glo1 protein. The S-nitrosylation is also associated with a strong inhibition of Glo1 activity [82,83,84]. Another PTM of Glo1 includes acetylation, which leads to ubiquitin-driven degradation of the enzyme, for instance in patients with non-alcoholic fatty liver disease [85]. Additionally, glutathionylation of Glo1 has been described, which strongly inhibited Glo1 activity in vitro, but the physiological relevance of this PTM remains elusive [84].

The most studied PTM of Glo1 is the phosphorylation. There are five different putative phosphorylation sites identified for Glo1, whereas Thr^107^ is the only one which has been identified as the one with physiological consequences [6,84]. The first description of the phosphorylation of Thr^107^ was in a fibrosarcome cell line after the exposure to TNFα, which is necessary in order to induce the expression of NFκB and subsequent cell death [86]. A recent study was able to shed more light on the molecular consequences of the phosphorylation of Thr^107^. It was revealed by a pharmacological and genetic approach that the driving kinase is Ca^2+^/Calmodulin-dependent Kinase II delta (CamKIIδ) [87]. Moreover, phosphorylated Glo1 has an elevated catalytic efficiency (lower K_m_; higher V_max_) and it is also protected from rapid proteasomal degradation by ubiquitination [87]. Given the background that other studies have shown that an acetylation enforces the proteasomal degradation, it is a valid hypothesis to claim that the phosphoryl group could prevent the acetylation of Glo1 in a competitive way [85]. Interestingly, the reduced state of Glo1-phosphorylation, and therefore reduced Glo1 activity in vitro and in vivo, was associated with only mild changes regarding the phenotype. Only MG-specific DNA adducts (MG-dG) were increased and this was linked to increased nuclear damage markers such as p53 and γH2Ax. This effect was more pronounced in isolated murine endothelial cells as compared to the in vivo mouse model [87]. An important finding of this recent study was the association of phosphorylated Glo1 with diabetes, ageing and malignant cells. In fact, decreased Glo1 phosphorylation has been associated with a decline in Glo1 activity in diabetes or ageing, whereas the opposite effect seems to take place in human tumor cells, in which Glo1 activity is usually upregulated [87]. This could provide an explanation for the altered Glo1 status in different intracellular environments, described by many other studies, which are mentioned in the introduction of this review [7,15,16,17,27,28,33,34,41]. Although these recent findings regarding the driving kinase of a Glo1 phosphorylation only scratch the surface of the underlying mechanisms, it might help to clarify the precise role of the glyoxalase system and Glo1 in particular. At this point it cannot be excluded that there are other kinases too, besides CamKIIδ, which are able to phosphorylate Glo1 efficiently. For instance, it has been shown that TNFα can induce multiple phosphorylation events at Glo1 [88].

The PTM of an enzyme is usually a quick and precise way to adapt in a flexible way to various intracellular conditions [89]. Nowadays, it is not fully understood why and how CamKIIδ is induced and why the phosphorylation of Glo1 is really needed. Keeping in mind that the solely claimed role of Glo1 is MG detoxification, one has to declare that Glo1 indeed has the highest efficiency in detoxifying MG as compared to alternative enzymes. AKR and especially ALDH are ten and one hundred times less efficient compared to Glo1 when it comes to their capacity and turnover-rate, even when the whole AKR/ALDH family detoxifies MG simultaneously [61,62,74]. This is also shown by the hypersensitivity towards exogenous MG in Glo1 KO in vitro models [64,72]. However, the capacity of AKR and ALDH is enough to prevent any major damage in a Glo1 KO mice and zebrafish model, even under hyperglycemic conditions [70,74]. This, therefore, brings up the question of whether the whole story of the glyoxalase system is already known? In the 1960s, the noble prize award winner Szent-Györgyi was interested in the fundamental role of the glyoxalase system, because of its highly conserved role and abundancy. He and his co-workers found that MG was inhibiting cell proliferation by interfering with protein synthesis [90]. Back then, they introduced the concept of promine (enhancer of cell proliferation) and retine (inhibitor of cell proliferation) and the captivating hypothesis that MG is the retin and Glo1 must be then the promine [91]. A concept which he was, unfortunately, unable to prove, but given the new insights into the glyoxalase system and especially the new background of a quick and precise regulation via phosphorylation of Glo1, it may be the right time to re-consider his idea [92].

## 4. Alternative Physiological Functions of Methylglyoxal

As pointed out at the beginning of this review, the intracellular concentrations of MG must be maintained at low levels in order to prevent cellular damage [5,7,16,17,18]. Although it has been speculated for some years that MG has important intracellular roles, there is little support for the idea that the production of MG in eukaryotes is a regulated process. An MG synthase has only been found in microorganisms supporting this concept, but no similar structures have been described in eukaryotes or even mammalian cells [93,94].

However, recent studies draw a slightly different picture. Sub-toxic doses of MG seem to be linked to a beneficial effect called hormesis. The concept of hormesis was first introduced by the pharmacologist Hugo Schulz, who observed that the growth rate of yeast could be stimulated by small doses of toxins [95]. Examples of hormetic effects in humans and animals are ionizing radiation at chronic low doses or the quite famous example of the benefits of low doses of alcohol consumption to reduce the risk of heart diseases and stroke [96,97]. Interestingly, the benefit of physical exercises and the associated positive effects in aging driven by the increase in reactive oxygen species (ROS), may display the strongest “hormetin” available in the human body. There is more and more evidence that repetitive mild stress exposure has an anti-ageing effect. Unfortunately, this “good” free-radical theory contradicts the common belief that the consumption of antioxidant supplements or eating of excessive amounts of fruits and vegetables may lead to a long and healthy life. Intervention studies that used Vitamin E or C showed no effect or even detrimental effects regarding antioxidative capacity [97,98]. As pointed out, the concept of the so called “mitohormesis” is able to challenge this common belief, with studies showing that the mitochondria need ROS in order to adapt and to prevent long-term overload of ROS under certain circumstances [99]. Metaphorically, one can phrase it as a “ROS training”, which occurs due to physical exercises or also starvation in the mitochondria and this is eventually a life-extending and health-promoting event.

Undoubtedly, MG increases ROS even at low levels, because of its high reactivity towards amino acids and nucleotides [77]. It modulates the intracellular environment, but does it also do that in an adaptive, training-like way? Looking into plant physiology, it has been shown that MG is involved in the closure of stomata by modulating ROS production in *Arabidopsis thaliana* [81]. In yeast it has been shown that low doses of MG induce a tolerance towards toxic concentrations of MG and H_2_O_2_ [100]. Interestingly, this MG-triggered tolerance is independent of Glo1 and secondly the finding could be translated into a mammalian in vitro system, in murine endothelial cells [100]. The effect is mostly driven by an increased expression of the protein quality system, in fact many heat shock proteins (HSPs) are upregulated, but also oxidoreductases, such as AKR (comparable to Glo1 KO in mammals), are increased due to the preconditioning with low doses of MG [100]. Going higher on the evolutionary ladder, an interesting study by Ristow et al. was performed in C. elegans. By interfering with the threonine catabolism, the formation of MG was promoted, which unexpectedly increased lifespan in worms [101]. It is a highly provocative study, since other studies have reported that an overexpression of Glo1, which reduced MG levels, prolongs lifespan in C. elegans [36]. In contrast, the knock-down of Glo1 in C. elegans revealed a decrease in lifespan and that was linked to elevated MG as well as MG-H1 levels [102]. Further studies are mandatory in order to reveal the true metabolic role of Glo1 and MG in C. elegans. Last but not least, when looking again into the first Glo1 KO in *Drosophila* fly, which recapitulates the progression of type 2 diabetes, the authors expected reduced longevity. Instead of that, it was reported that the Glo1 KO and potentially the mild increase in MG are associated with a significant lifespan extension [71]. This result was confirmed by the study of Ristow et al. in the model organism C. elegans and suggests an evolutionary conserved role for this mechanism [101].

The nature of a biphasic dose–response relationship (hormesis) has been assessed in human tumor cell lines as well. Hormetic-like relationships in malignant cells have been reported for endogenous agonist (e.g., hormones in breast cancer cells), antitumor drugs (e.g., cisplatin in cervix cancer cells) or phytochemicals (e.g., resveratrol in leukemia) [103]. The hormetic potential of MG in cancer cells has been discovered recently in different breast cancer and glioblastoma cell lines. In this study, the maintenance of tolerable MG stress at subtoxic levels was beneficial regarding cancer growth and resistance to apoptosis. Such responses seem to be mainly driven by the upregulation of Nrf2 [104]. Another recent study in a very aggressive and lethal form of thyroid cancer reported comparable effects. In this in vitro approach, it was not MG itself that was responsible for the hormetic effect, but the accumulation of MG adducts that turned out to enhance the migration and invasion of anaplastic thyroid cancer. Interestingly, using resveratrol, which has been proven as an inducer of Glo1, the hormetic effect of increased MG adducts was lost [105]. The glycation of HSPs by MG in cancer cells, such as HSP27 or HSP90, could explain partly the metabolic advantage of cancer cells, which are reflected by a resistance towards cisplatin [15]. Investigating hormetic effects in cancer cells is a new field of research, which should be investigated thoroughly, in order to understand the interplay of MG and the glyoxalase system better in highly proliferating cells.

With respect to MG adducts, there is another example of an unexpected experimental result within the glyoxalase metabolism. Besides MG-H1, which accounts for more than 90% of MG protein modifications, there is also the possibility that MG-H3 is formed [21] (Figure 1). Although it is not clear to what extent MG-H3 occurs physiologically and how stable this compound is, an in vitro study revealed that MG-H3 was able to act as an antioxidant with a comparable efficiency to ascorbic acid in scavenging free radicals. It would be of great interest if this compound could function as a bulk antioxidant within a broader physiological context [106].

## 5. Conclusions

This review showed that the glyoxalase system is surprisingly not an indispensable system to maintain cellular viability. Especially in mammalian organisms, the loss of the glyoxalase system is compensated efficiently by enzymes of the AKR and ALDH family. Those enzymes can play a pivotal role within the context of MG mediated damage in healthy cells or malignant cells. Undoubtedly, and this has been shown herein, there is room for studying the glyoxalase metabolism from alternative points of view. This includes PTMs of Glo1, hormetic effects of MG and MG-derived AGEs, or even the intriguing concept that the glyoxalase system may reflect a regulator of cell division. It is a pity that many studies have focused relatively blindly on the cause-and-effect model between Glo1 and MG, which has been described in detail in the introduction of this review. This is further supported by the idea that there is so little literature published on Glo2 in the last 10 years (123 hits for Glo1 vs. eight hits for Glo2; [107]). This review wants to polarize and may provoke a bit in order to emphasize and encourage all researchers in the field to look beyond the obvious and investigate the glyoxalase system without prejudice.

## Figures and Tables

**Figure 1 antioxidants-09-00939-f001:**
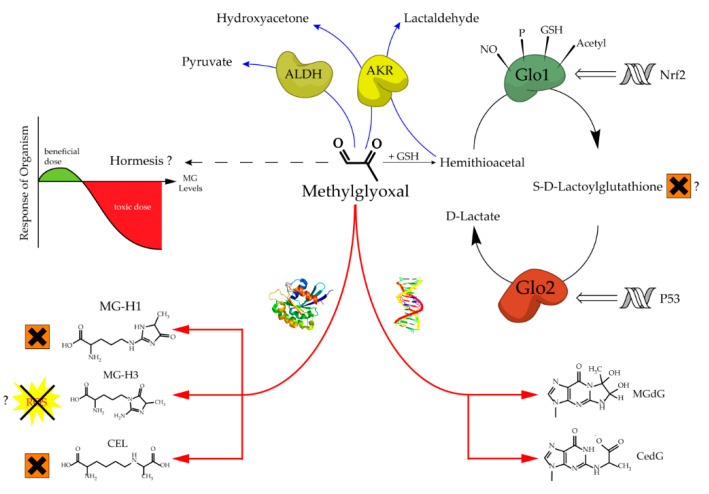
Aspects of methylglyoxal metabolism and its possible implications.

**Figure 2 antioxidants-09-00939-f002:**
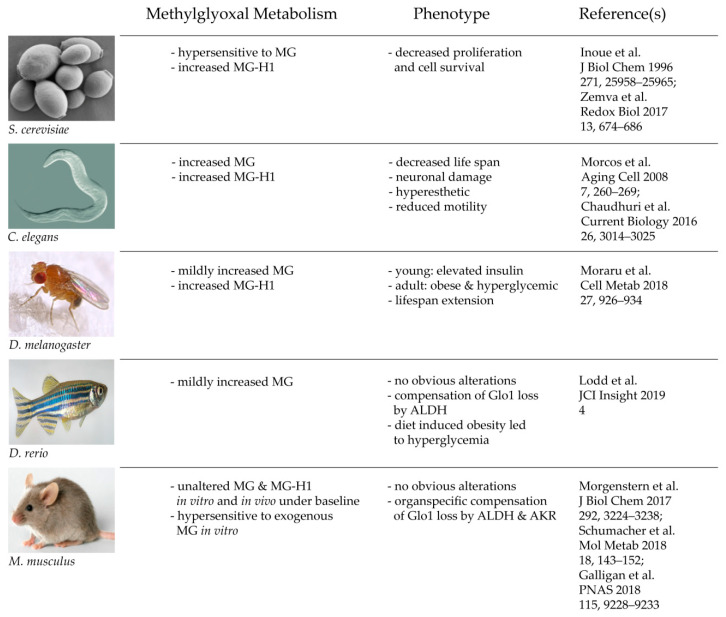
Consequences of a total loss of glyoxalase 1 in various model organisms.

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
