# Peer review of "The Glyoxalase System—New Insights into an Ancient Metabolism"

_antioxidants, 2020, doi:10.3390/antiox9100939_

Round 1

Reviewer 1 Report

This is an excellent critical appraisal of the data published in the MG-GLO-1 pathway. Although several aspects of glyoxalase 1 and methylglyoxal stress were recently addressed in a comprehensive paper of Schalkwijk en Stehouwer in Physiological Review (2020), the present manuscript pay special attention to unexpected findings about  glyoxalase 1 and the degradation of MG and the  potential beneficial effects of MG under physiological conditions.  It is worthwhile and in fact essential to bring these aspects under attention of the readers of Antioxidants and the scientific society, instead of ignoring the unexpected findings based on the assumption that there are only harmful aspects of this pathway. That there are only harmful effects is definitely not true and the authors very nicely summarized and described that in this review.

Some point to consider:

  1. Line 69; In my view it is not obvious from literature that MG-H1 accounts for 90% of all AGEs.
  2. Line 73; CeDG (CdG!) is the major MG-modified deoxyguanosine adduct.
  3. Line 86: Because of compensation mechanisms of the degradation of MG in Glo-1 KO mice by ALDH and by AKR and therefore the uncertainty to draw conclusions about Glo-1, it should be mentioned that Glo-1 overexpression reduces microvascular complications in diabetic animal models. Because of all kind of compensation mechanisms for Glo-1 in Glo-1 KO, it looks that the data in Glo-1 overexpression animals are much more informative about the function of Glo-1.  
  4. Line 121 Figure 1: Is clear, but the word and symbol “toxic” for MG and MG-derived AGEs are indicating a dangerous and extreme condition and that is not. I suggest to change toxic for harmful. It might be that there are also important physiological aspects of MG-H1 and CEL, such as for MG.
  5. Line 272: ref 80 is definitely an important paper, but in press. Waiting for a release on short term to get knowledge with the details.

Author Response

The authors would like to thank the reviewer for providing us with constructive criticism and suggestions to improve the quality of this manuscript.

Reviewer #1 (Specific Comments):

  1. Line 69; In my view it is not obvious from literature that MG-H1 accounts for 90% of all AGEs.

Response:

The authors agree that the original finding is not provided by the cited literature. Therefore, supporting literature has been added and the sentence is rephrased to the following: "Based upon quantitative measurements, MG-H1 is the dominant type of all AGEs found within a physiological context" (Line 69-70).

  1. Line 73; CeDG (CdG!) is the major MG-modified deoxyguanosine adduct.

Response:

The authors changed the abbreviation accordingly also in the right bottom of Figure 1 and in the list of abbreviations (Line 73-75, Line 121-23 & Line 423).

  1. Line 86: Because of compensation mechanisms of the degradation of MG in Glo-1 KO mice by ALDH and by AKR and therefore the uncertainty to draw conclusions about Glo-1, it should be mentioned that Glo-1 overexpression reduces microvascular complications in diabetic animal models. Because of all kind of compensation mechanisms for Glo-1 in Glo-1 KO, it looks that the data in Glo-1 overexpression animals are much more informative about the function of Glo-1.

Response:

The authors agree that there are several studies published regarding the benefits of an overexpression of Glo1 in diabetic animal models. Therefore, an additional sentence and two references have been added to this section (Line 88-90).

  1. Line 121 Figure 1: Is clear, but the word and symbol “toxic” for MG and MG-derived AGEs are indicating a dangerous and extreme condition and that is not. I suggest to change toxic for harmful. It might be that there are also important physiological aspects of MG-H1 and CEL, such as for MG.

Response:

The authors agree given also the potential hormetic and therefore, beneficial effect of MG and MG derived AGEs, it is more appropriate to have a less dangerous imagery. The symbol "toxic" was changed to "harmful" (Line 121-123).

  1. Line 272: ref 80 is definitely an important paper, but in press. Waiting for a release on short term to get knowledge with the details.

Response:

The authors now added the full citation due to the final publication recently (Line 281).

Reviewer 2 Report

In general the review is well-written and contains all the relevant literature on the field. Minor issues are highlighted in the pdf attached file. 

Author Response

The authors would like to thank the reviewer for thoroughly reading our manuscript and providing us with constructive criticism.

  1. In E. Coli, recent discovery have been proposed a single step detoxification of MG into D-lactate by a unique glyoxalase pathway (glyoxalase III) without the need of any cofactor. Please refer to: Sankaranarayanan S., Jamshed M., Kumar A., Skori L., Scandola S., Wang T., Spiegel D., SamuelInt D.A. Glyoxalase Goes Green: The Expanding Roles of Glyoxalase in Plants. J. Mol. Sci. 2017, 18, 898

Response:

The authors thank the reviewer for this valuable literature, but this fact is more suitable in the second part, within the context of other/alternative mechanisms of detoxification. Therefore, the following sentence was added: Interestingly, in Escherichia coli it has been shown that there is also a glyoxalase-like enzyme (glyoxalase 3) which is independent of any cofactor, converting MG into D-Lactate. The article, which was identified in the provided literature, is now cited accordingly (Line 131-133).

  1. In fact, in plants glyoxalases exist as a multigene family and they have been largely studied from a physiological perspective, being linked to stress tolerance in plants. I suggest to increase the overview about plants. There are several papers about the topic. As example the following: Charanpreet Kaur, Sneh L. Singla-Pareek & Sudhir K. Sopory (2014); Glyoxalase and Methylglyoxal as Biomarkers for Plant Stress Tolerance, Critical Reviews in Plant Sciences, 33:6, 429-456; Proietti, S.; Falconieri, G.S.; Bertini, L.; Baccelli, I.; Paccosi, E.; Belardo, A.; Timperio, A.M.; Caruso, C. GLYI4 Plays A Role in Methylglyoxal Detoxification and Jasmonate-Mediated Stress Responses in Arabidopsis thaliana. Biomolecules 2019, 9, 635

Response:

The authors expanded the role of glyoxalases in plants and added further comments regarding glyoxalases and cell division, tolerance to abiotic stresses, role in the process of pollination, and Glo induction towards stress related hormones (Line 258-265).

  1. Also in plants, glyoxalases undergo phosphorylation. In rice leaf, different GLYI phosphorylations have been found in response to low temperatures. Please refers to: Charanpreet Kaur, Sneh L. Singla-Pareek & Sudhir K. Sopory (2014) Glyoxalase and Methylglyoxal as Biomarkers for Plant Stress Tolerance, Critical Reviews in Plant Sciences, 33:6, 429-456

Response:

The authors added a description about gibberelin as a regulator of growth, which also induces phosphorylation of Glo1 (Line 258-259).

  1. Several studies have been demonstrated that MG acts as signalling molecule in plants. In Arabidopsis, it has been proved that MG involves the closure of stomata by modulating ROS production. Please refer to: Proietti, S.; Falconieri, G.S.; Bertini, L.; Baccelli, I.; Paccosi, E.; Belardo, A.; Timperio, A.M.; Caruso, C. GLYI4 Plays A Role in Methylglyoxal Detoxification and Jasmonate-Mediated Stress Responses in Arabidopsis thaliana. Biomolecules 2019, 9, 635 Sankaranarayanan S., Jamshed M., Kumar A., Skori L., Scandola S., Wang T., Spiegel D., SamuelInt D.A. Glyoxalase Goes Green: The Expanding Roles of Glyoxalase in Plants. J. Mol. Sci. 2017, 18, 898

Response:

The authors added a short sentence about this intriguing finding in plants. "Looking into plant physiology, it has been shown that MG is involved in the closure of stomata by modulating ROS production in Arabidopsis thaliana." (Line 348-349).